# Risk Assessment and Application of Tea Frost Hazard in Hangzhou City Based on the Random Forest Algorithm

**Ying Han [1], Yongjian He [1,\*], Zhuoran Liang [2], Guoping Shi [1], Xiaochen Zhu [3] and Xinfa Qiu [3]**

1   School of Geographical Sciences, Nanjing University of Information Science & Technology, Nanjing 210044, China
2   Hangzhou Meteorological Bureau, Hangzhou 310000, China
3   School of Applied Meteorology, Nanjing University of Information Science and Technology, Nanjing 210044, China
*   Correspondence: 001529@nuist.edu.cn

**Abstract:** Using traditional tea frost hazard risk assessment results as sample data, the four indicators of minimum temperature, altitude, tea planting area, and tea yield were selected to consider the risk of hazard-causing factors, the exposure of hazard-bearing bodies, and the vulnerability of hazard-bearing bodies. The random forest algorithm was used to construct the frost hazard risk assessment model of Hangzhou tea, and hazard risk assessment was carried out on tea with different cold resistances in Hangzhou. The model's accuracy reached 93% after training, and the interpretation reached more than 0.937. According to the risk assessment results of tea with different cold resistance, the high-risk areas of weak cold resistance tea were the most, followed by medium cold resistance and the least strong cold resistance. Compared with the traditional method, the prediction result of the random forest model has a deviation of only 1.57%. Using the random forest model to replace the artificial setting of the weight factor in the traditional method has the advantages of simple operation, high time efficiency, and high result accuracy. The prediction results have been verified by the existing hazard data. The model conforms to the actual situation and has certain guiding for local agricultural production and early warning of hazards.

**Keywords:** random forest; machine learning; GIS; tea frost hazard; hazard risk assessment; hazard warning

## 1. Introduction

Frost is the main hazard affecting the quality of tea, especially in early spring, and has the greatest impact on tea [1–3]. There are many hills and mountains in Hangzhou City, Zhejiang Province, coupled with a monsoon climate, which makes the Hangzhou area an excellent environment for tea production [4,5]. Tea is an economically important crop in Hangzhou. According to the 2019 Statistical Yearbook of Hangzhou, tea production in Hangzhou was 31,342 tons, the planting area was 35,255 hectares, and the total output value accounted for 33% of the agricultural output value. However, a cold wave in April 2019 caused more than 4000 acres of tea damage in Hangzhou, which hindered the tea picking process and caused serious losses to tea farmers. Therefore, it is important to strengthen research concerning the frost hazard risk assessment of tea, formulate hazard prevention and mitigation countermeasures, and reduce economic losses [6–13].

In recent years, machine learning has been applied to various fields. The earliest machine learning appeared in the 1970s, but limited by the computer level at that time, there were no particularly good applications, and development progress was slow. At the beginning of the 21st century, machine learning began a period of vigorous development, and random forest (RF) was produced during this period [14]. At present, many scholars at home and abroad use machine learning algorithms, such as random forests, for hazard risk assessment, mainly focused on hazards such as typhoons, rainstorms, floods, and

landslides [15–18]. Domestic and foreign studies on hazard risk zoning using the random forest algorithm and other machine learning methods were mainly applied in mountain torrents, landslides, and floods after 2010 [19–22]. Lai Chengguang et al. [23] compared decision trees, support vector machines, particle swarms, and artificial neural networks and found that random forests have a good effect on multivariate prediction and can be applied to many fields. Li Ting et al. [24] applied the random forest method to landslide hazard zoning and analyzed the importance and influential factors. Wu Xiaojun et al. [25] chose the random forest algorithm to establish a mountain torrent hazard risk assessment model in Jiangxi Province, divide the risk levels, and draw a mountain torrent hazard risk zoning map in Jiangxi Province. Chen Junfei et al. [26] compared the results of the support vector machine and random forest algorithms in flood hazard risk assessment. They found that the accuracy of flood hazard risk assessment based on random forest was higher than that of the support vector machine model. Zhou Chao et al. [27] analyzed the risk of flash flood hazards in Jiangxi Province by using three algorithms, k-nearest neighbor, random forest, and AdaBoost, and found that the accuracy and Kappa coefficient of AdaBoostd were slightly better than those of random forest, while the three performance indicators of the k-nearest neighbor model were all lower than those of the former two algorithms.

In the risk assessment of single-crop meteorological hazards, conventional risk assessment methods were mostly used [28–32], and machine learning algorithms, such as random forest, were rarely used. Therefore, based on the random forest algorithm, this study established a risk assessment model for tea frost hazards in Hangzhou. This model can effectively overcome the subjectivity of weight in traditional assessment methods and can quickly obtain the required data in the model. The hazard risk assessment results of the study area change over time in the later period. The relevant departments consistently update the data to obtain new hazard risk assessment results, which provide a reliable method and basis for the current tea frost hazard prediction. This study has important reference meaning for early warning of tea frost hazards and provides ideas for related research.

## 2. Materials and Methods

### 2.1. Research Area

Hangzhou is located in the middle and lower reaches of the Yangtze River, with many hills and suitable temperatures. It is an ideal place for a tea plantation, and they are famous for Longjing tea, Jiukeng Maojian, Qiandao Yinzhen, etc. However, in March and April every year, the Hangzhou area often encounters frost hazards, which bring huge economic losses to local tea farmers. Although there is a relatively accurate weather forecast system, most can only forecast one or two days in advance. Tea farmers can only rush to harvest part of the yield, and there is still nothing to do about the losses caused by the final hazard. Therefore, it is of great significance to carry out a risk assessment of frost hazards in various regions of Hangzhou and determine the high-risk areas of tea frost hazards for the planning of planting in the agricultural sector and the prevention of key hazards.

### 2.2. Data and Data Processing

#### 2.2.1. Data Sources

The data used in this study mainly included meteorological station data, altitude data, and statistical yearbook data. Considering the time consistency, this paper used the observation data of 7 conventional meteorological observation stations and 80 automatic meteorological observation stations (Figure 1) in Hangzhou. The observation time ran from March 2000 to September 2019. The elevation data used DEM data with a resolution of 100 m released by NASA in 2017; the tea-related data came from the 2019 Hangzhou Statistical Yearbook released by the Hangzhou Statistics Bureau.

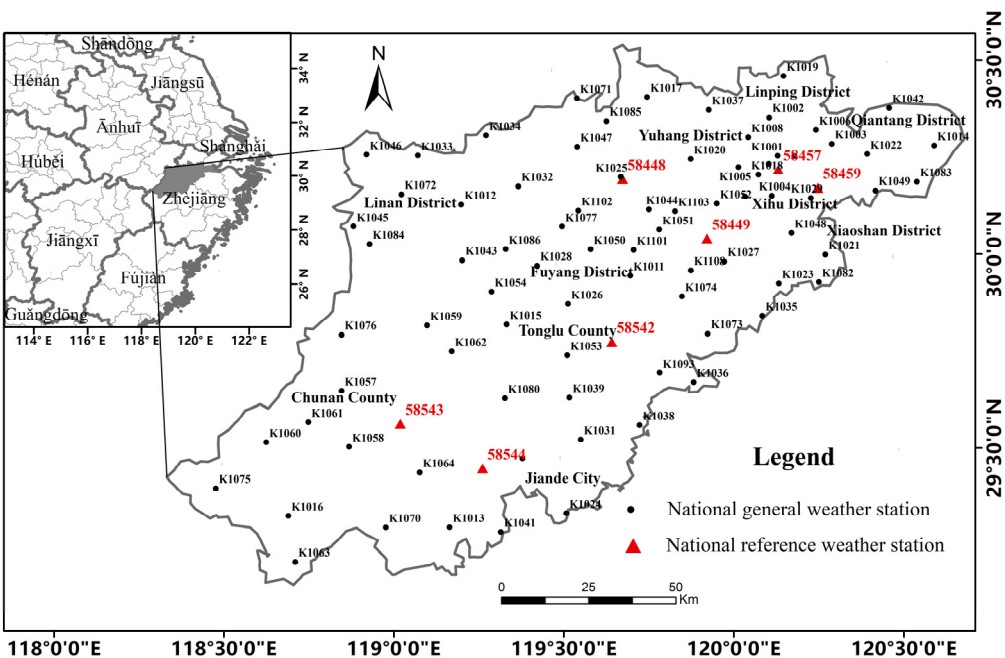

**Figure 1.** Distribution map of weather stations.

### 2.2.2. Selection of Key Factors

According to risk hazard assessment theory, six kinds of data, including temperature data, humidity data, wind speed data, elevation data, tea planting area, and yield data, were selected as the main factors. The analytic hierarchy process was used to further determine the key factors. According to the analytic hierarchy process (AHP), a judgment matrix was first constructed to judge the relative importance of each standard to measure whether the goal can be achieved and screen six kinds of data. $a_i$, $a_j$ (i, j = 1,2...n) represent the elements being compared. aij represents the relative importance value of ai to aj, which is given in the following by Santy [33]. According to the literature, the number of factors for pairwise comparison should not exceed 9; that is, each layer should not exceed 9 factors [34,35]. The elements in the judgment matrix were compared with the scale, and the judgment matrix was obtained to represent the measurement of the importance of each element. The scale of the judgment matrix (Table 1) and the actual judgment matrix (Table 2) are shown in the table below.

**Table 1.** Judgment matrix scale comparison table.

| Scaling | Meaning |
|---------|---------|
| 1 | Indicates that two factors have the same importance compared to |
| 3 | Indicates that when compared to two factors, one factor is slightly more important than the other |
| 5 | Indicates that when two factors are compared, one factor is significantly more important than the other |
| 7 | Indicates that when two factors are compared, one factor is strongly more important than the other |
| 9 | Indicates that when two factors are compared, one factor is extremely more important than the other |
| 2, 4, 6, 8 | The median of the above judgments |
| reciprocal | Factors i and j compare and judge aij, then the comparison and judgment between factor j and i aji = 1/aij |

**Table 2.** The construction of judgment matrix results.

|  | Humidity | Wind Speed | Tea Production | Tea Area | Altitude | Low Temperature |
|---|---|---|---|---|---|---|
| Humidity | 1 | 0.667 | 0.5 | 0.5 | 0.25 | 0.2 |
| Wind speed | 1.5 | 1 | 0.667 | 0.5 | 0.33 | 0.25 |
| Tea production | 2 | 1.5 | 1 | 0.667 | 0.5 | 0.33 |
| Tea area | 2 | 2 | 1.5 | 1 | 0.5 | 0.33 |
| Altitude | 4 | 3 | 2 | 2 | 1 | 0.5 |
| Low temperature | 5 | 4 | 3 | 3 | 2 | 1 |

Then, according to the judgment matrix, the eigenvector $\omega$ corresponding to the largest eigenvalue $\lambda_{max}$ is obtained, and the equation is as follows:

$$P \, \omega = \lambda_{max} \, \omega \tag{1}$$

The obtained feature vector was normalized so that the sum of each element in the vector was equal to 1, and the weight value of each evaluation factor was obtained, as shown in Table 3. The calculation results of the AHP show that the largest characteristic root is 6.045, the corresponding CI value is 0.009, and the RI value is 1.26, according to the RI table, so CR = CI/RI = 0.007 < 0.1, it means that the judgment matrix of this study satisfies the consistency test, and the calculated weights are consistent.

**Table 3.** Tea frost factor weight.

| Factor | Low Temperature | Altitude | Tea Area | Tea Production | Wind Speed | Humidity |
|---|---|---|---|---|---|---|
| weights | 0.367 | 0.234 | 0.138 | 0.115 | 0.081 | 0.065 |

According to the factor weight results, low-temperature data were selected as the final hazard-causing factor, altitude data were selected for the vulnerability factors of hazard-bearing bodies, and tea area and yield data were selected for the exposure of hazard-bearing bodies.

2.2.3. Classification of Tea Leaves with Differences in Cold Resistance

According to the characteristics of the tea itself, the tea was divided into weak cold resistance, medium cold resistance, and strong cold resistance [36], and the frost hazard risks of tea with different cold resistances were calculated. According to previous studies on frost hazards of tea leaves in Hangzhou [13,37–46], different hazard-causing factor indicators were set for tea with different cold resistances. The specific indicators are shown in the following table (Table 4). The risk of frost hazards for different cold-resistant tea leaves was spatialized (Figure 2). The data used the WGS1984 projection coordinate system, and the spatial resolution of the data was 100 × 100 m. According to the indicators of different cold-resistant tea leaves in Table 4, statistical data from meteorological stations were used to obtain the disaster-causing factors of tea leaves with different cold resistance; the resulting data was interpolated and then normalized.

**Table 4.** Indicators of tea with different cold resistance in Hangzhou.

| Types of Tea | Indicator Period | Indicator Data |
|---|---|---|
| weak cold resistance tea | February–May | Number of days with daily minimum temperature $\leq 4\,^{\circ}\mathrm{C}$ |
| medium cold resistance tea | February–May | Number of days with daily minimum temperature $\leq 0\,^{\circ}\mathrm{C}$ |
| strong cold resistance tea | February–May | Number of days with daily minimum temperature $\leq -2\,^{\circ}\mathrm{C}$ |

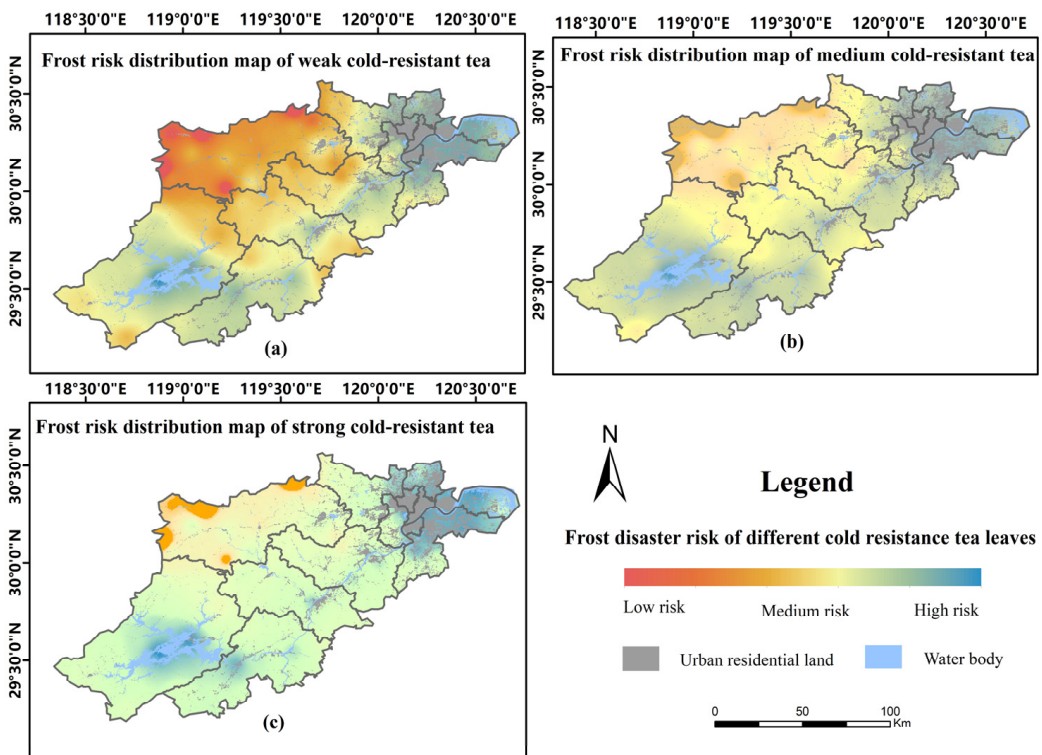

**Figure 2.** Spatial distribution of model indicators: (**a**) Spatial distribution map of early spring frost-induced disaster factors for weak cold-resistant tea; (**b**) Spatial distribution map of early-spring frost-induced disaster factors for moderate cold-resistant tea; (**c**) Spatial distribution map of early spring frost-induced disaster factors for strong cold-resistant tea.

### 2.3. Methods

#### 2.3.1. Traditional Meteorological Hazard Risk Assessment Model

Presently, research on natural hazard risk mainly focuses on three aspects: the formation mechanism of hazard risk, the method of hazard risk assessment, and the technology of hazard risk assessment [47]. According to the latest risk expression proposed by the IPCC Fifth Assessment Report:

$$Risk = W_h \times H \times W_e \times E \times W_v \cdot V \tag{2}$$

where *Risk* is the hazard risk index, H, E, and V represent the risk of hazard-causing factors, the exposure of hazard-bearing bodies, and the vulnerability factors of hazard-bearing bodies, respectively, and $W_h$, $W_e$, and $W_v$ are the risk of hazard-causing factors, the exposure of hazard-bearing bodies, and the vulnerability factors of hazard-bearing body weights, respectively. In this paper, based on basic geographic, meteorological, and socioeconomic data, a meteorological hazard risk assessment model based on evaluation factors was established in accordance with the risk of hazards, exposure of hazard-affected bodies, and vulnerability of hazard-affected bodies. Risk assessment research was carried out, and GIS technology was used to draw the hazard risk zoning map of the research area.

Traditional meteorological hazard risk assessment mainly adopts the analytic hierarchy process, which mainly relies on an expert scoring form to determine the different importance of each element, which is highly subjective. When the elements change, expert scoring needs to be performed again. In addition, after experts score, when each element changes due to external conditions, its importance will also change accordingly. If this method is to be used, it needs to be re-evaluated. In this study, the random forest algorithm was used to replace the subjective and cumbersome weighting process in the traditional method, which can quickly and conveniently evaluate meteorological hazards with less deviation in the evaluation results.

2.3.2. Random Forest Algorithm

The random forest algorithm is a machine learning algorithm based on statistics and combination classification. It has strong nonlinear simulation ability and generalization. Because fewer parameters are artificially set, the subjectivity of evaluating research questions can be reduced [48,49]. Its basic principle is a collection of bagging algorithms and random subspace algorithms. The basic unit is a decision tree. Multiple decision trees are combined to form a forest. Each decision tree is used to classify and predict votes to obtain the final classification and evaluation results.

The random forest uses the bagging algorithm to sample the dataset. The probability of being drawn for each sample in the total training set D with a sample size of N is $(1 - 1/N)^N$. When the N area is infinite, the result is 0.368. This means that more than 1/3 of the data will not appear in the extracted dataset, and these data are called out-of-bag (OOB). Using these out-of-bag data, the misclassification rate of the classification tree is calculated to obtain the OOB error, and the average of the OOB errors of all decision trees in the model can be used as the generalization error of the model to illustrate the generalization ability of the model. The method of calculating the importance of risk indicators is reducing the Gini index when the node is divided, and the sum of the Gini coefficients of all nodes in the forest is averaged for all trees. The specific formula is as follows (Formula (3)):

$$P_k = \frac{\sum_{i=1}^{n} \sum_{j=1}^{t} D_{Gkij}}{\sum_{k=1}^{m} \sum_{i=1}^{n} \sum_{j=1}^{t} D_{Gkij}} \times 100\% \tag{3}$$

In the formula, $m$, $n$, and $t$ are the total number of indicators, the number of classification trees, and the number of nodes of a single tree, respectively; $D_{Gkij}$ is the Gini index reduction value of the $k$ indicator in the $j$ node of the tree; and $P_k$ is the $k$ indicator in all indicators' degrees of importance.

*2.4. Overall Flow Chart*

The flow chart of the technical approach in this paper is shown in Figure 3. It includes data preparation, key feature selection, and random forest model building and validation. Data preparation includes acquiring and processing meteorological, terrain, land use, and tea-related data. The key feature selection is to select six kinds of data that may cause disasters according to the principle of tea frost and obtain the most important feature data through the AHP method. Model building and verification use the random forest algorithm to divide, sample, verify, and test the data to obtain the evaluation model of this study, and then input the data to be tested to obtain the tea disaster risk assessment results. The evaluation results are compared, verified, and analyzed through the measured disasters in the Disaster Statistical Yearbook.

*2.5. Construction of the Tea Frost Assessment Model in Hangzhou*

2.5.1. Sample Selection

The selection of samples is crucial to the evaluation accuracy of random forest, and the quality determines the final accuracy and overall effect. Most of the records of hazards by relevant departments and literature are for a certain type of hazard, but there are few records for a single hazard or a single crop, and they are not detailed. Through data search, there are only a dozen records and descriptions of frost hazard records on tea in Hangzhou, which is not enough to support the input data requirements of the random forest model. However, Lai Chengguang and others divided the Dongjiang River Basin into five flood hazard levels. Points were taken evenly on them as samples. Therefore, based on previous studies and expert opinions, this study uses the tea frost risk data in Hangzhou obtained from traditional hazard risk theory (taking tea with weak cold resistance as an example) (Figure 4) as a basis to extract sample point data.

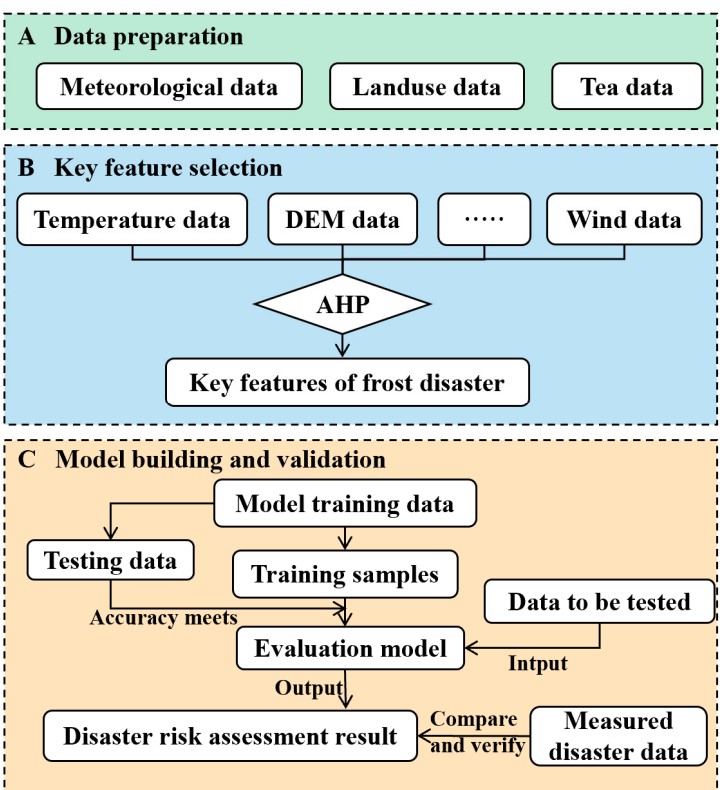

**Figure 3.** Technical method flow chart.

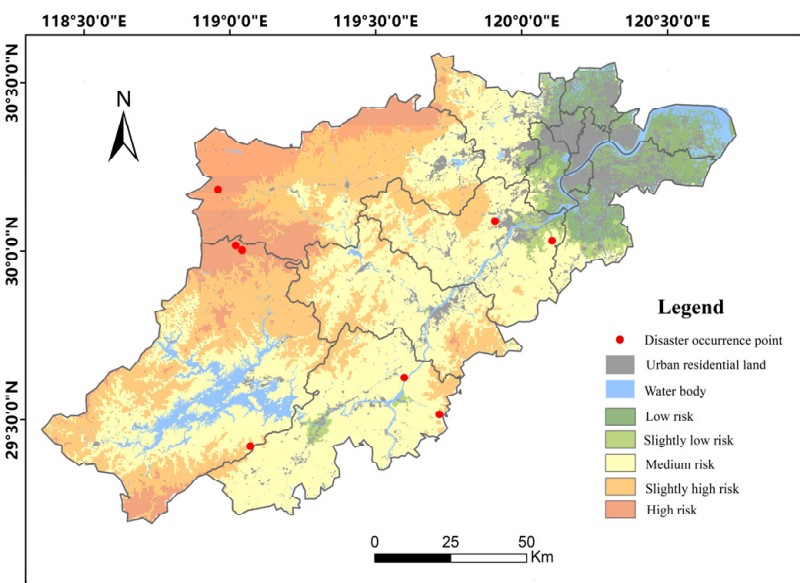

**Figure 4.** Risk map of freezing injury of tea with weak cold resistance.

According to the hazard annual report data in the basic database of hazard prevention and reduction in Hangzhou, the locations of the actual tea frost hazards are all in areas with high-risk levels (Figure 4), which proves that it has high reliability and can be used as a source for model sample selection. In Hangzhou water bodies and other areas outside urban residential land, 1000 sample point data points were evenly selected, and the specific spatial distribution is shown in Figure 5 below.

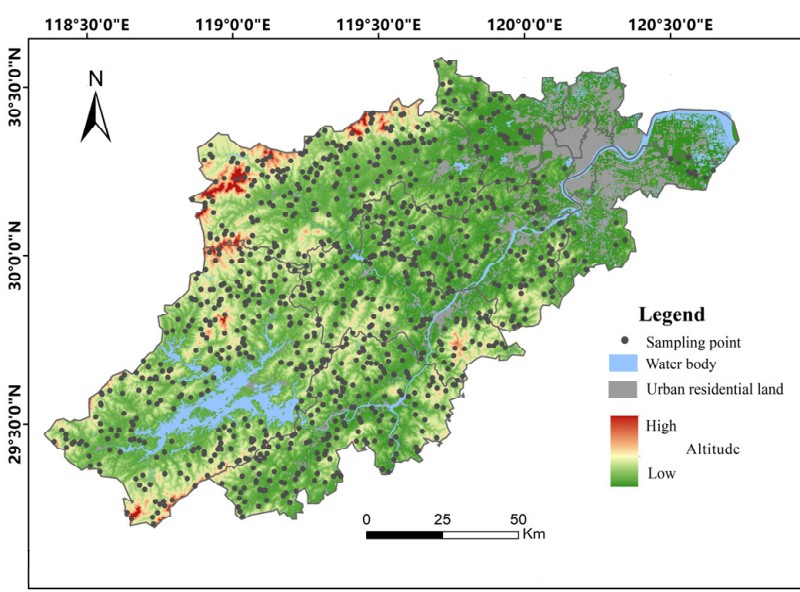

**Figure 5.** Sample point distribution map.

### 2.5.2. Model Construction

Compared with other algorithms, the random forest algorithm has better performance, can better handle multi-feature data, and does not need to perform feature selection. After data processing and sample data selection, the Hangzhou tea frost hazard risk model was constructed in the software, the sample data were read and divided first, and the data samples were divided into a training set and a testing set. The ratio of the training set to the test set was 4:1, the training set was used to generate the tea frost model, and the testing set was used to test the final prediction results. Then, the grid search function (RandomizedSearchCV) in the Sklearn library was used to obtain the optimal parameters. If the accuracy requirements are met, the next calculation can be performed. If the accuracy requirements are not met, resampling and parameter setting are needed. After obtaining the optimal parameters, input the training data and validation data for model training and construction. The evaluation parameters were obtained through model training, as shown in the following table (Table 5). The interpretation degrees of the training set and the testing set reached 0.991 and 0.959, respectively, indicating that the model interprets the data very well. The OOB score of the generalization ability of the weak cold resistance model was 0.9699, indicating that the model had good accuracy, robustness, and generalization ability. The OBB scores of the medium and strong cold-resistant tea models are 0.9495 and 0.9378, respectively, and both have strong accuracy and generalization ability.

**Table 5.** Model training result evaluation.

| Evaluation Standard | Training Set | Testing Set |
|:---:|:---:|:---:|
| $R^2$ | 0.991 | 0.9597 |
| MSE | 0.00005 | 0.0048 |
| Absolute difference | 0.0048 | 0.0101 |
| Explainability | 0.991 | 0.9597 |

Finally, all the index data used for frost hazard risk are input into the trained random forest model. The predicted risk assessment results are input to obtain the regional map of tea frost risk assessment in Hangzhou. During the calculation process regarding the random forest model, the importance of each index was calculated according to the average Gini reduction value of all nodes, and the importance of each index to the formation of frost hazard risk was calculated. The specific data are shown in Figure 6 below.

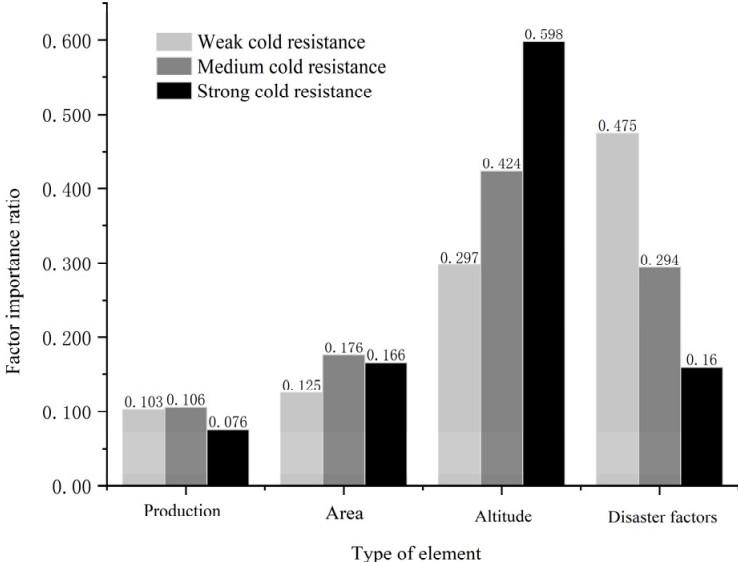

**Figure 6.** Element Importance Distribution.

From the perspective of different types of cold-resistant tea, the most important indicator of weak cold-resistant tea is altitude. The most important indicator of medium and strong cold-resistant tea is the hazard-causing factor composed of low-temperature data. Due to the weak cold resistance and temperature drop rate of tea with weak cold resistance, altitude is the main factor for the risk of frost hazards, while tea with medium cold resistance and strong cold resistance have strong cold resistance. To overcome the influence of altitude, the main influencing factor is the hazard data composed of the lowest temperature. Overall, hazard-causing factors and altitude data are the most important contributing indicators. The sum of the two is close to 70%, indicating that it contributes the most to the risk assessment results. The influence of altitude and hazard-causing factors should be considered when planting tea trees.

## 3. Results

### 3.1. Results of Frost Hazard Based on Random Forest Tea Leaves

A total of 51,772 pieces of data from the four indicators of tea frost in Hangzhou that were screened out were input into the training model to obtain the current frost hazard risk prediction value of tea with different cold resistance. The results were classified according to the natural breakpoint method, and the current tea frost hazard risk prediction and assessment map was obtained (Figure 7).

It can be seen from the resulting map (Figure 6) that the overall high-risk areas of tea frost hazard in Hangzhou are mainly distributed in Lin'an District, Jiande City, and the Fuyang District, and the higher-risk areas are distributed in Chun'an County, Lin'an District, and Tonglu County. The low-risk areas are mainly distributed in the plain areas of Linping District, Qiantang District, and Shangcheng District, and the remaining areas are medium-risk areas. The risks to tea with different cold resistance are also different, and the high-risk areas of tea with weak cold resistance and medium cold resistance are mainly concentrated in mountainous areas with higher altitudes, while the risk of strong cold-resistant tea is lower in mountainous areas with higher altitudes, and the ability to resist risks is stronger. Table 6 shows that the high-risk area of frost hazard is highest for tea with weak cold resistance, reaching 10%, followed by tea with medium cold resistance, and tea with strong cold resistance is the lowest. Overall, weak cold resistance and medium cold resistance teas accounted for the largest proportion of medium risk and high risk, while strong cold resistance teas accounted for the largest proportion of medium risk and low risk.

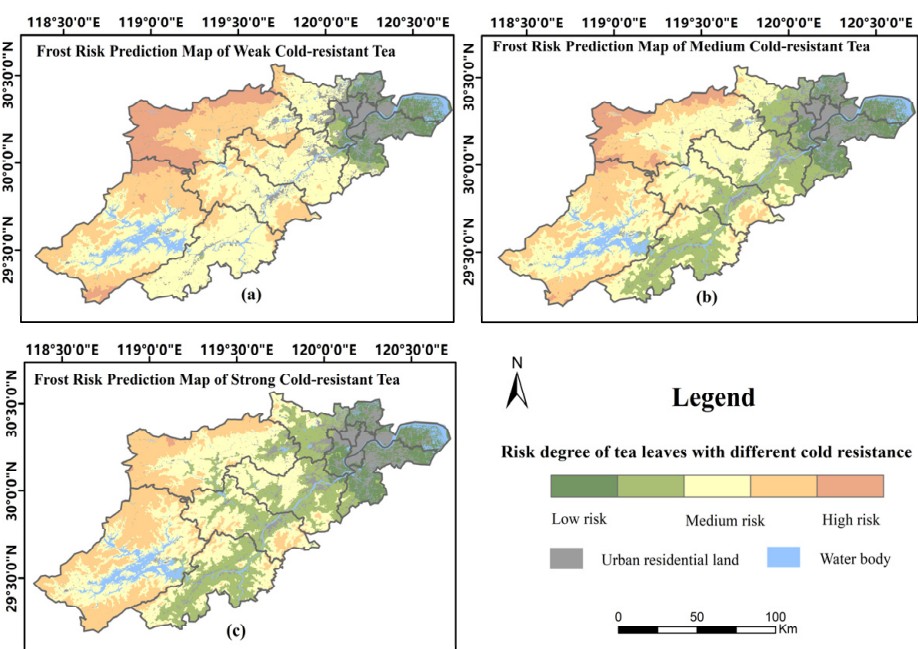

**Figure 7.** Frost disaster risk map of tea with different cold resistance: (**a**) Prediction of freezing injury risk of tea with weak cold resistance; (**b**) Prediction of risk of freezing injury of tea with moderate cold resistance; (**c**) Prediction of risk of freezing injury of tea with strong cold resistance.

**Table 6.** The proportion of frost disaster results in teas with different cold resistance.

|  | Weak Cold Resistance | Medium Cold Resistance | Weak Cold Resistance |
| --- | --- | --- | --- |
| low risk | 11% | 12% | 14% |
| slightly low risk | 3% | 22% | 26% |
| medium risk | 48% | 41% | 37% |
| slightly risk | 28% | 20% | 22% |
| high risk | 10% | 5% | 2% |

*3.2. Analysis of Tea Frost Hazard Results in Hangzhou*

Figure 5 shows the altitude of Hangzhou City. The north and west of Lin'an District are high-altitude mountainous areas easily affected by frost hazards. The value of hazard-causing factors is also high, so the risk level of this area is also high risk. The high-risk areas in the north of Jiande City and the northwest of Fuyang District are mainly due to the high value of the hazard-causing factors. Through the analysis of the importance of factors, it can be seen that the hazard-causing factors have the largest proportion of importance and play a decisive role in the high risks of these areas. The high-risk areas and higher-risk areas in Chun'an County are mainly distributed in areas with higher altitudes. The second reason is the planting area of tea. The tea planting area of Chun'an County in 2019 was 12,404 hectares, far exceeding the second planting area of Yuhang District of 4264 hectares. However, due to the low terrain, many plains, and densely populated areas in the eastern part of Hangzhou City, most of the plain areas are mostly artificial land and residential land. Most tea plantations are in mountainous and hilly areas. Factors, such as area and tea production, have minor values, so in the final risk assessment, most are in the low-risk area.

By randomly selecting 800 comparison points, the frost hazard risk value and the corresponding values of altitude and hazard-causing factors were extracted for analysis (Figure 8). The first is the correlation between the hazard-causing factor and the risk value of the frost hazard. The correlation coefficient between the two is the highest, reaching 0.65, which aligns with the main principle of the tea frost hazard. To protect tea from the tea frost hazard, the most important thing is to pay attention to the cooling of the cold wave and

the altitude. The correlation coefficient between the two reaches 0.5187, which has a high correlation, indicating that the final frost hazard risk result has a strong correlation with the altitude factor. It is also consistent with the fact that altitude is more important when building the model. Therefore, farmers need to avoid high-altitude areas when actually planting tea and choose areas with suitable altitudes for planting.

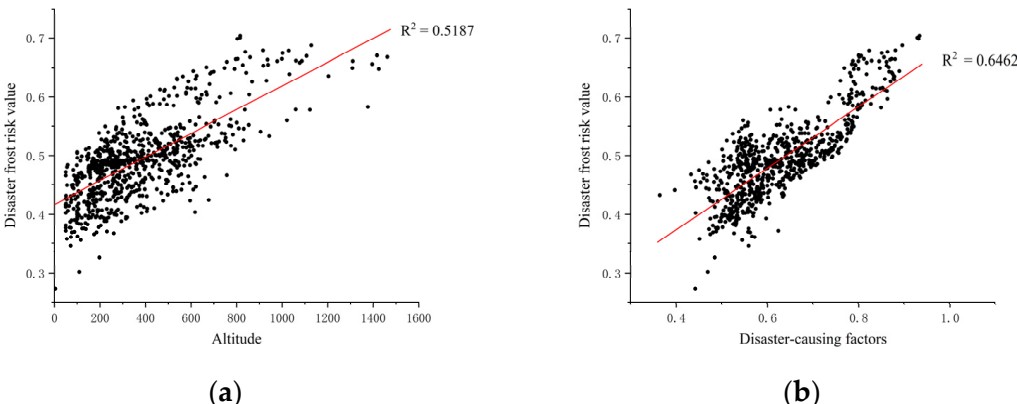

**Figure 8.** Correlation between frost disaster risk value and different factors: (**a**) Frost hazard risk value and altitude correlation; (**b**) The correlation between frost disaster risk value and disaster-causing factors.

### 3.3. Verification of Tea Frost Hazard Results in Hangzhou

According to the location and affected area of the tea frost hazard in the Hangzhou Disaster Prevention and Mitigation Basic Database, a spatial display was carried out on the frost risk zoning of tea in Hangzhou (taking tea with weak cold resistance as an example) (Figure 9). The change from grey to red indicates an increase in the severity of the hazard, and the size of the area indicates the affected area. To facilitate the comparison of the tea frost hazard risk results, the hazard data were set to 30% transparency. By extracting the data of actual hazard occurrence points and comparing the results of the traditional method and the random forest model method, the deviation between the two methods was only 1.57%. The overall risk distribution of the two methods is consistent, only slightly different in mountainous areas, which further verifies the reliability of the random forest model.

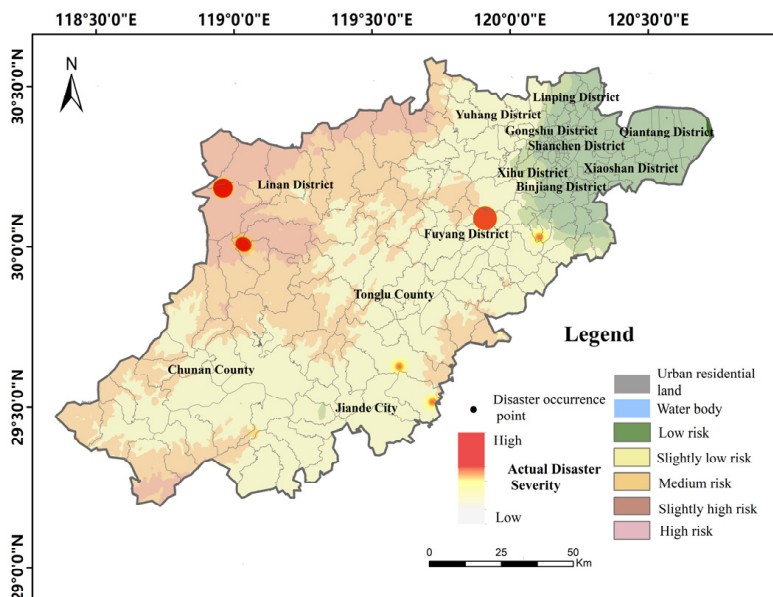

**Figure 9.** Frozen injury of tea.

It can be seen from Figure 9 that the most serious events of tea frost hazards are mainly in Qingliangfeng Township of Lin'an District, Yaoshan Township of Chun'an County, Fuchun Street of Fuyang District, Qiantan Township and Sandu Township of Jiande Township, while the high-risk and slightly high-risk areas are consistent with the locations where tea frost hazards actually occurred. Individual frost hazards with fewer hazards are in the medium-risk area, indicating that the assessment of the evaluation results is highly reliable and scientific. The hazard risk distribution of the result map is also in line with the actual observation data. It has a high degree of interpretation, indicating that the trained random forest model meets expectations for tea frost prediction, which can provide a basis and support for relevant departments and personnel.

## 4. Discussion and Conclusions

This study is mainly based on the hazard risk assessment results based on the hazard risk theory proposed by the IPCC, from which data samples were selected, and the random forest algorithm in machine learning was used as the main tool to construct a comprehensive analysis of the tea frost hazard in Hangzhou. Based on the risk assessment model, the following conclusions were obtained:

1. The hazard assessment model constructed using random forest does not need to set index weights and classification standards in advance. It is easy to operate, has a small amount of code, and is easy to modify and maintain later. Only the input data must be updated, and the optimal parameters must be adjusted to obtain the prediction results closest to the current situation, reducing user burden and time. Relevant departments can quickly obtain future disaster assessment results and then formulate targeted disaster prevention and mitigation policies to reduce losses caused by disasters;
2. The accuracy and interpretation of the model reached 94%, which is a high standard. The prediction results also align with the actual situation of hazard statistics. The deviation from the traditional results is only 1.57%, which can be used as an important basis for relevant departments to respond to hazards;
3. The importance of each element to the final result can be analyzed from the factor importance obtained by the model, which can make more accurate judgments on analysis and evaluation and more accurate preparations for hazard prevention;
4. The method steps of this study can be extended to more crops and hazards and provide a reference for related research. Therefore, relevant departments can respond to different disasters faster and better.

Finally, there are several improvements and shortcomings in this study:

1. More elemental data can be used to improve the accuracy and scientificity of the results;
2. There are few hazard records for crops. If there are enough relevant records available in the future, the results will be more accurate.

**Author Contributions:** Conceptualization, Y.H. (Ying Han)., Y.H. (Yongjian He), G.S., X.Z. and X.Q.; methodology, Y.H. (Ying Han); software, Y.H. (Ying Han); validation, Y.H. (Ying Han). and Y.H. (Yongjian He); formal analysis, Y.H. (Ying Han); investigation, Y.H. (Ying Han); resources, Y.H. (Ying Han); data curation, Y.H., Z.L. and X.Z.; writing—original draft preparation, Y.H. (Ying Han); writing—review and editing, Y.H. (Ying Han), Y.H. (Yongjian He), and X.Q.; visualization, Y.H. (Ying Han); supervision, Y.H. (Ying Han); project administration, Y.H. (Ying Han) and Y.H. (Yongjian He); funding acquisition, Y.H. (Ying Han) and Y.H. (Yongjian He). All authors have read and agreed to the published version of the manuscript.

**Funding:** This research was funded by the National Natural Science Foundation of China (41971295) and the 2021 Jiangsu Province Postgraduate Research and Practice Innovation Plan Project (KYCX21_0934).

**Institutional Review Board Statement:** Not applicable.

**Data Availability Statement:** The data presented in this study are available on request from the corresponding author. The data are not publicly available due to the privacy policy of the authors' institution.

**Acknowledgments:** Funding from the National Natural Science Foundation of China (41971298) and the 2021 Jiangsu Province Postgraduate Research and Practice Innovation Plan Project (KYCX21_0934) is gratefully acknowledged. The authors gratefully acknowledge the Hangzhou Meteorological Bureau for offering the raw data. We also thank the editors and reviewers for their comments to improve our manuscript.

**Conflicts of Interest:** The authors declare no conflict of interest.

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
