# Peer review of "Risk Assessment and Application of Tea Frost Hazard in Hangzhou City Based on the Random Forest Algorithm"

_agriculture, doi:10.3390/agriculture13020327_

Round 1

Reviewer 1 Report

This paper uses machine learning to explore the risk assessment of tea plantation which is an important economic crop in Asia. With the threat of climate change, if risk assessment can provide early warning to plantation planning, it can provide better economic security to the rural communities. 

Having emphasized the above research background and importance, the last part of this paper - the concluding discussion can be further improved by linking the technical analysis with future policy recommendations. This last part should be of importance and can be added to increase the practical application and contribution of this study. 

Author Response

Point 1: Having emphasized the above research background and importance, the last part of this paper - the concluding discussion can be further improved by linking the technical analysis with future policy recommendations. This last part should be of importance and can be added to increase the practical application and contribution of this study.  

Response 1: Thanks for your suggestions on this article. Based on your suggestion, I have added more relevant suggestions and analysis in the final summary and discussion section, illustrating the practical significance and broad application of this research. Finally, thank you again for your valuable advice.

Reviewer 2 Report

The manuscript threats with an interesting topic in the context of climate change and uses topical methodologies that can provide valuable results.

The paper starts with a good review of the scientific background and a sufficient number of references cited. The manuscript is well-written, clear, and precise, with figures and tables that strongly support the manuscript`s message.

In the Introduction part, I suggest discussing the novelty of the analysis and the gap in the knowledge to be addressed by this paper. Also, in this chapter, reference can be made to the study’s objectives.

Please, in figure 1, represent the stations more understandably. Also, within this map, you can make a map with the area’s location.

Line 113-114 Please cite Saaty's methodology.

Line 113 – please replace using Santy's 1-9 scale with Saaty, and the authors can cite in the text this methodological article.

The results obtained are checked by calculating the Consistency Ratio (CR) please add in the text the result obtained and the matrix. These are more important than Table 1.

It needs to be clarified how frost risk distribution maps are made (Figure 2). Please add more methodological details.

Line 191 The authors are referring to figure 3. Please check.

Line 208-210 – please check

Figure 6. Please make the graph in a more representative way

The authors can represent the districts on the map ~distributed in Lin'an District, Jiande City and 297 Fuyang District, and the higher-risk areas are distributed in Chun'an County, Lin'an Dis-298 trict and Tonglu County~ because it's quite difficult to track the results.

In the discussion part, a comparative analysis of the results obtained in the two analysis methods can be made and more discussions based on the results obtained.

Author Response

Point 1: Please, in figure 1, represent the stations more understandably. Also, within this map, you can make a map with the area’s location.

Response 1: I have used different colors to distinguish weather stations of different grades, but I have not yet found a suitable icon to represent a weather station. A map of southeast China has been added for a better understanding of the location of the study area.

Point 2: Line 113-114 Please cite Saaty's methodology;Line 113 – please replace using Santy's 1-9 scale with Saaty, and the authors can cite in the text this methodological article;The results obtained are checked by calculating the Consistency Ratio (CR) please add in the text the result obtained and the matrix. These are more important than Table 1.

Response 2: I've added a reference to Saaty, a table of the actual confusion matrix and the calculated CR values after Table 1.

Point 3: It needs to be clarified how frost risk distribution maps are made (Figure 2). Please add more methodological details.

Response 3: I have added a detailed explanation of the relevant content.

Point 4: Line 191 The authors are referring to figure 3. Please check;Line 208-210 – please check.

Response 4: I have modified the content or layout of the formatting error.

Point 5: Figure 6. Please make the graph in a more representative way.

Response 5: I have replaced the different elements in the histogram with a more easily distinguishable format.

Point 6: The authors can represent the districts on the map ~distributed in Lin'an District, Jiande City and 297 Fuyang District, and the higher-risk areas are distributed in Chun'an County, Lin'an Dis-298 trict and Tonglu County~ because it's quite difficult to track the results.In the discussion part, a comparative analysis of the results obtained in the two analysis methods can be made and more discussions based on the results obtained.

Response 6: I have added the names of the specific districts and counties of Hangzhou City in the picture to better understand the specific location. Added more analysis where the two different approaches are discussed. Finally, thank you again for your valuable advice.

Reviewer 3 Report

The most important drawback and weakness of this article is its introduction and literature. The introduction was very poorly written. The innovation was not very clear. The use of RF which was proposed 20 years ago is not highly innovative in itself. The importance of the subject (tea frost hazard) was not well expressed. Previous studies in this field have not been well reviewed. Some sentences need reference. "At the beginning of the 21st century, machine learning began a period of vigorous development, and random forest (RF) was born during this period." The structure of the paper was not suitable and it should be improved. The conclusion was necessary for this paper. Figure 3 was not needed for the paper. Instead, a schematic flowchart or graphical abstract of the methodology of the paper is necessary for the paper. A comparative analysis among some algorithms is needed to enrich the paper. In general, the current form of this paper is not appropriate for this journal.

Author Response

Point 1: The introduction was very poorly written. The innovation was not very clear. The use of RF which was proposed 20 years ago is not highly innovative in itself. The importance of the subject (tea frost hazard) was not well expressed.

Response 1: Thank you for your comments. The innovations of this article are mainly in the abstract and the third paragraph of the introduction. The random forest method is the result of selection based on the operability of the research and the actual application effect, and other more novel methods can be considered in the future. The importance of tea disasters is mainly reflected in the first paragraph of the introduction.

Point 2: Some sentences need reference."At the beginning of the 21st century, machine learning began a period of vigorous development, and random forest (RF) was born during this period."

Response 2: I have added relevant reference.

Point 3: Figure 3 was not needed for the paper. Instead, a schematic flowchart or graphical abstract of the methodology of the paper is necessary for the paper.

Response 3: Figure 3 is not only the flow chart of random forest, but also the flow chart of the main idea of this paper.

Point 4: A comparative analysis among some algorithms is needed to enrich the paper.

Response 4: At this stage, this paper only studies the random forest algorithm and compares it with the traditional method. In the follow-up research, I will add different algorithms for correlation analysis and comparison to enrich the research. Finally, thank you again for your valuable advice.

Round 2

Reviewer 2 Report

Thanks for the changes. 

I recommend a careful check of the manuscript (e.g. as I mentioned earlier, to correct Santy with Saaty) the same error now appears in the bibliography. 

After verification, I agree to publish the article.

Author Response

Point 1: improve figure 3 to make it more understandable and to allow a better readability of the paper.  

Response 1: Referring to the relevant literature recommended by the editor, I have improve Figure 3 into a technical flow chart, and added a new paragraph to it, briefly introducing the main content of the flow chart, so that readers can better read this article.

Point 2: Rename the last paragraph as "Discussion and conclusions"

Response 2: I have revised the paragraph title.

Point 3: Revise English throughout the paper (e.g. "According to the location and affected area of the frost hazard involving tea in the hazard annual report in the basic database of hazard prevention and mitigation in Hangzhou, spatial display was carried out on the frost risk zoning of tea in Hangzhou (taking tea with weak cold resistance as an example)" repeats hazard 3 times in one sentence).

Response 3: I have read through paper, revise the sentences and words that have reading problems, and set the unit words in italics, about 10 modifications. You can open the "revision model" to see the specific modification content.

Reviewer 3 Report

Unfortunately, almost none of the comments have been answered well by the authors.

Some my comments were deleted from the Author_response file!

"Previous studies in this field have not been well reviewed."

"The structure of the paper was not suitable and it should be improved. The conclusion was necessary for this paper."

 This is a kind of disrespect to the reviewer!

Author Response

(The authors gave the same response as above.)
